# Dynamic Response of Pile-Slab Retaining Wall Structure under Rockfall Impact

Peng Zou[1, 2, 3], Gang Luo[1, *], Yuzhang Bi[4], Hanhua Xu[2, 3]

1. *Faculty of Geosciences and Engineering, Southwest Jiaotong University, Chengdu 611756, China*

2. *Kunming Prospecting Design Institute of China Nonferrous Metals Industry Co., Ltd, Yunnan 650051, Chin*a

3. *Yunnan Key Laboratory of Geotechnical Engineering and Geohazards, Kunming, 650051, China*

4. *College of Resources and Environment, Fujian Agriculture and Forestry University, Fuzhou 350002, China*

**Corresponding author at:** Faculty of Geosciences and Engineering, Southwest Jiaotong University, Chengdu 611756, China

E-mail addresses: luogang@home.swjtu.edu.cn (G. Luo).

**Abstract:** The pile-slab retaining wall, as an innovative rockfall protection structure, has been extensively utilized in the western mountainous regions of China. With its characteristics of a small footprint, high interception height, and ease of construction, this structure demonstrates promising potential for application in mountainous regions worldwide, such as the Himalayas, Andes, and Alps. However, its dynamic response upon impact and impact resistance energy remain ambiguous, due to the intricate composite nature of the structure. To elucidate this, an exhaustive dynamic analysis of a four-span pile-slab retaining wall with a cantilever section of 6 m under various impact scenarios was conducted utilizing the finite element numerical simulation method. The rationality of the selected material constitutive models and the numerical algorithm was validated by reproducing two physical model tests. The simulation results reveal the following: (1) The lateral displacement of the pile at the ground surface and the concrete damage under the pile as the impact center is greater than those under the slab as the impact center, implying that the impact location has a significant influence on the stability of the structure. (2) There is a positive correlation between the response indexes (impact force, interaction force, lateral deformation of pile and slab, concrete damage) and the impact velocities. (3) The rockfall peak impact force, the ratio of peak impact force to peak interaction force, and lateral displacement of pile at the ground surface had strong linear relationships with rockfall energy. (4) Relative to the bending moment, shear force and damage degree, the lateral displacement of pile at the ground surface is the first to reach its limit value. Taking the lateral displacement of the pile at the ground surface as the controlling factor, the estimated maximum impact energy that the pile-slab retaining wall can withstand is 905 kJ in this study when the structure top is taken as the impact point. In cases where the impact energy of falling rocks exceeds 905 kJ, it is recommended to optimize the mechanical properties of the cushion layer, improve the elastic modulus of concrete, increase the reinforcement ratio of longitudinal tension bars, enlarge the section size of pile at ground level, or add anchoring measures to enhance the bending resistance of the retaining structure.

**Keywords:** rockfall, pile-slab retaining wall, numerical simulation, dynamic response

## List of symbols

| | | | | |
|---|---|---|---|---|
| $P$ | Actual lateral soil resistance (kPa). | | $F_{dm}$ | Peak impact force (kN). |
| $P_u$ | Ultimate lateral soil resistance (kPa). | | $F_{im}$ | Peak interaction force (kN). |
| $S_{u\_cu}$ | Consolidated isotropic undrained triaxial shear strength of soil (kPa/m). | | $\alpha$ | Ratio of the peak impact force to the peak interaction force (%). |
| $y$ | Actual lateral soil deformation (m). | | $S_{mpt}$ | Maximum lateral displacement of pile at the ground surface (mm). |
| $B$ | Pile width (m). | | $N_d$ | Number of damage failure units. |
| $z$ | Depth below the ground surface (m). | | $\beta$ | Ratio of damage failure units to overall structure units (%). |
| $S_p$ | Shape correction factor of pile section. | | $m$ | Impactor mass (kg). |
| $E$ | Initial kinetic energy of impactor. | | $v$ | Initial velocity of impactor (m/s). |

## 1. Introduction

Rockfall disasters pose a great threat to roads, railways, buildings and inhabitants in mountainous terrain (Hungr et al., 2014; Crosta and Agliardi, 2004; Shen et al., 2019). It can be described as a process that the rapid bouncing, rolling and sliding movement of one (or several) boulders down a slope (Peila and Ronco, 2009). Muraishi et al. (2005) surveyed 607 rockfall events and found that about 68% of rockfall events have an impact energy of less than 100 kJ, whereas 90% have less than 1000 kJ. Chau et al. (2002) indicated that the rotational kinetic energy of rockfall only accounts for 10% of the total kinetic energy. To mitigate such geological hazards, scholars and engineers have proposed different types of technical solutions. Two primary categories of defensive measures are commonly employed: active and passive. Active protection measures mainly include masonry protection, reinforcement protection (grouting, anchor rod, and anchor cable), initiative protective net (Yang et al., 2019). Passive protection measures include passive flexible protection (Yu et al., 2021), rockfall shed gallery (Zhao et al., 2018), rockfall retaining wall. Considering many factors, such as technological feasibility and economic considerations, rockfall retaining wall is frequently employed in practical engineering (Volkwein et al., 2011).

Currently, various types of retaining walls are utilized in engineering projects aimed at intercepting falling boulders. These include masonry retaining walls, reinforced concrete (RC) retaining walls, reinforced soil retaining walls, and pile-slab retaining walls (PSRW). Due to

inherent structural weakness of these walls, their ability to absorb the impact energy from rockfall is limited (Mavrouli et al., 2017). To enhance the impact resistance, the reinforced concrete retaining walls have been utilized (Yong et al., 2020). These structures can intercept rockfall impact energy ranging approximately from 120 to 500 kJ (Maegawa et al., 2011). To prevent concrete from being damaged by the direct impact of rockfall, a buffer layer is generally added in front of the structure for protection, such as reinforced soil and gabion cushion (Perera et al., 2021). Although the impact resistance of the structure has been improved, there is still a problem of limited interception height. When the required interception height is large, the foundation size has to be increased to prevent the structures from overturning. In order to mitigate against rockfall events involving higher energy levels, numerous researchers have proposed the implementation of reinforced soil retaining walls. Extensive studies have been conducted in this regard, demonstrating that the structures can effectively intercept rockfall impact energies exceeding 5000 kJ (Lambert et al., 2009). Moreover, geosynthetic have proven to be efficacious in reducing wall stresses (Lu et al., 2021). However, the structure requires a substantial spatial footprint and poses an overturning risk during construction in steep terrain (Peila et al., 2007). Additionally, when the topography at the wall site features steep slopes, the available space behind the wall for accommodating rockfalls becomes constrained.

In response to the challenges posed by steep terrains, narrow site conditions, and suboptimal foundation conditions in mountainous terrain, Hu et al. (2019) introduced the PSRW structure. The structures are composed of a buffer layer and an anti-slid pile-slab structure, which has found widespread application in southwestern China (Fig. 1). Due to its implementation of pile foundations, this structure possesses characteristics such as a small footprint, high interception height, and ease of construction. However, the current PSRW design verification is to treat the structure as an underground continuous wall (CAGHP, 2019). And, due to the composite nature of this structure, the dynamical response at various impact points remains ambiguous. The maximum impact energy that the structure can withstand has also not been thoroughly investigated. It can lead to potential underestimation of failure possibilities (Fig. 1d). At the same time, the existing research focuses on the single slab and pile impacted by rockfall (Wu et al., 2021; Yong et al., 2021).

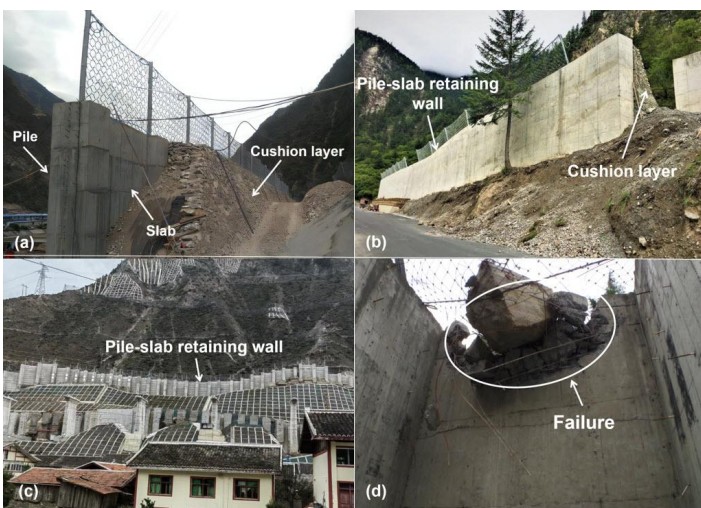

**Fig. 1.** PSRW in south-western China (a) Kongyu town (b) Jiuzhaigou nature reserve (c) Zhenjiangguan tunnel exit in Chengdu-lanzhou railway (d) Wenchuan-Maerkang expressway.

91        Therefore, analysis of structural dynamic response and concrete damage is crucial to

determine its effectiveness in mitigating rockfall hazards. Based on the unique advantages of the
finite element method, this study employs the LS-DYNA to simulate the complete process of
rockfall impacting on PSRW. This methodology has been widely adopted by numerous researchers
and demonstrated as suitable for simulating impact problems of reinforced concrete structure
(Zhong et al., 2022; Fan et al., 2022; Bi et al., 2023). In conclusion, a full-scale numerical model
of a four-span pile-slab retaining wall satisfying specification requirements is established. The
rationality of the selected material constitutive models and a numerical algorithm was validated by
reproducing two physical model tests. The structure's dynamic behavior under different impact
velocities and impact centers is discussed(Fig. 2). The results provide insights into sturcture
dynamic response analysis of the PSRW and serve as a benchmark for further research.

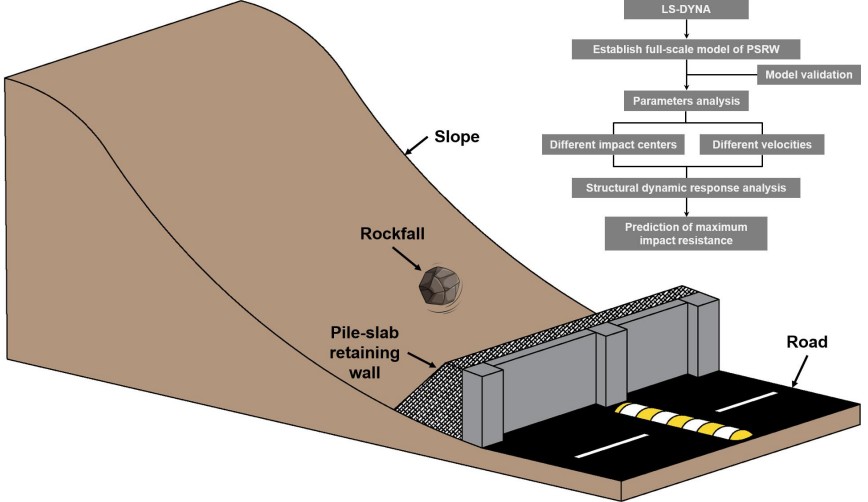

Fig. 2 Mind mapping.

## 2. Numerical model and validations

### 2.1. Model configuration

#### 2.1.1. Engineering background

The design drawing of the PSRW (Fig. 3) is consistent with the actual project located in Zhangmu Town, China. Given the large scale of the actual engineering structure, numerical simulations have been focused solely on a representative four-span structure, incorporating appropriately simplified boundary conditions to facilitate the analysis. For a comprehensive understanding of the modeling specifics, kindly refer to Section 2.1.3. The anti-slide piles with a concrete protective layer thickness of 0.04 m have a cross-section area of 1.8 m × 1.25 m. The total pile length is 12 m, and the embedded section is 6 m. The HRB 400 longitudinal bar with diameters of 25 mm and 32 mm were arranged in the pile (Fig. 3c). The stirrups are HRB335 with a diameter of 16 mm and a spacing of 200 mm. The slabs between the piles are 6 m in length, 3.5 m in width, and 0.5 m in thickness. These slabs contain two layers of 16 mm-diameter reinforced bar. The sand buffer layer are 1 m and 5 m on top and bottom, respectively. A geogrid is horizontally placed in the buffer layer at 0.25 m intervals. Lastly, 1 m³ sphere rock boulder with a diameter of 1.24 m was set as an impactor. The impact locations are 2# slab center (CS) and 3# pile center (CP) at 5.25 m over the ground.

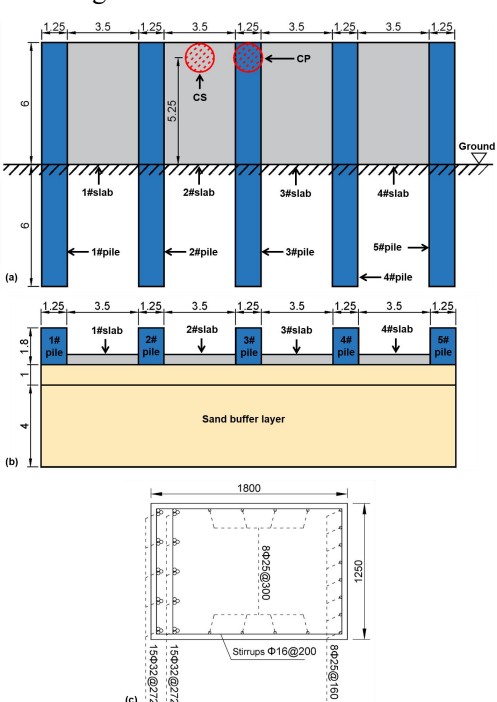

**Fig. 3.** The design diagram of PSRW (a) front view (unit: m) (b) top view (unit: m) (c) cross-section profile of pile (unit: mm).

*2.1.2.    Soil-pile interaction*
Under the impacting, the lateral deformations of the pile are greatly influenced by the plastic
behavior of the soil, particularly the soil near the pile. Given their importance and complexity, it
isn't easy to thoroughly describe soil-pile interactions. This paper calculates the pile-soil
interaction by the lateral resistance-deflection (*p-y*) curve method. As state by Truong and Lehane
(2018), the *p-y* curves for square cross-section pile are utilized as

$$\frac{P}{P_{\mathrm{u}}} = \tanh\left[5.45\left(\frac{y}{B}\right)^{0.52}\right] \tag{1}$$


$$\frac{P}{s_{\mathrm{u\_cu}}} = 10.5\left[1 - 0.75e^{-0.6z/B}\right]S_{\mathrm{p}} \tag{2}$$


where $P$ is the actual lateral soil resistance, kPa; $P_{\mathrm{u}}$ is the ultimate lateral soil resistance, kPa;
$S_{\mathrm{u\_cu}}$ is consolidated isotropic undrained triaxial shear strength of soil, kPa/m; $y$ is the actual lateral
soil deformation, m; $B$ is pile width, m; $z$ is depth below the soil surface, m; $S_{\mathrm{p}}$ is a shape
correction factor.
According to the reference and simulated model, the $S_{\mathrm{u\_cu}}$ and $S_{\mathrm{p}}$ are adopted as 1.5 kPa/m
and 1.25, respectively. Besides, the soil is modeled by compressive inelastic springs, arranged
every 0.25 m along the pile height and side (Fig. 4a).
*2.1.3.    Numerical model and numerical simulation scheme*
(1) Numerical model
The numerical model of PSRW is shown in Fig. 4. The material constitutive models, unit
types, physical-mechanical parameters, and parameter source for all components are listed in
Table 1. The rationality of all material constitutive models and physical mechanics parameters
were verified in Section 2.2. The bottom of piles and buffer layers are fixed for the boundary
conditions. Additionally, both sides of the buffer layer are blocked by infinitely rigid walls. The
contact type between the rockfall, sand buffer layer, and pile-slab structure was set to automatic
surface-to-surface.
(2) Numerical simulation scheme
According to previous research (Muraishi et al., 2005; Chau et al., 2002), angular velocity of
impactor was neglected in numerical simulations, and line velocities were set as 10, 15, 20, 25,
and 30 m/s, corresponding to impact energies of 130, 292.5, 520, 812.5, and 1170 kJ (Table 2).
The linear velocity is perpendicular to surface of the buffer layer.

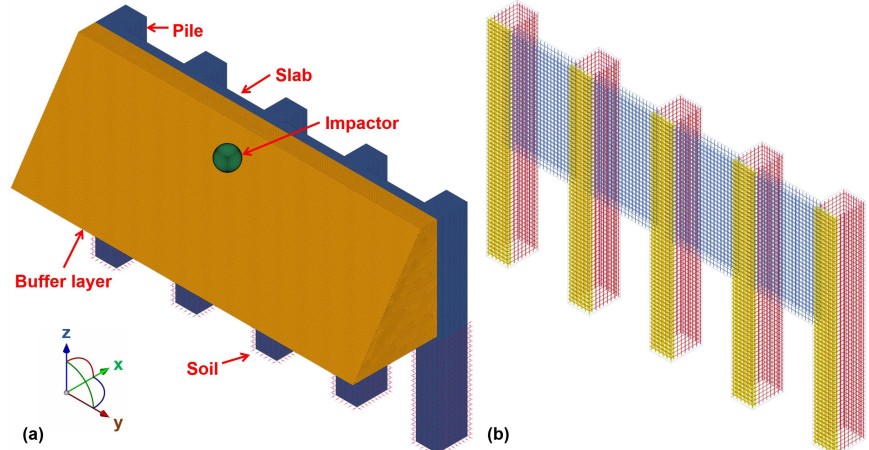

**Fig. 4.** Numerical model of the PSRW (a) numerical model (b) reinforced bar of PSRW (unit: mm).

**Table 1** Material constitutive model and physical-mechanical parameters for various components of PSRW.

| Items | Constrained model | Unit types | Integral methods | Density (kg/m³) | Young's modul (MPa) | Poisson's ratio |
|---|---|---|---|---|---|---|
| Concrete | Continue cap concrete (MAT_159) (Heng et al., 2021) | Solid element | One integration point | 2450 | 30000 | 0.3 |
| Reinforced bar | Plastic kinematic model (MAT_003) (Heng et al., 2021) | Beam element | 2×2 Gauss integration | 7850 | 204000 | 0.3 |
| Sand buffer layer | Soil-foam model (MAT_063) (Bhatti and Kishi, 2010) | Solid element | One integration point | 1720 | 100 | 0.3 |
| Impactor | Rigid body (MAT_020) | Solid element | One integration point | 2600 | 20000 | 0.25 |
| Geogrid | Plastic kinematic model (MAT_003) (Lee et al., 2010) | Shell element | Belytschko-Tsay integration | 1030 | 464 | 0.3 |

**Table 2** Detailed numerical simulation scheme.

| Case | Impact location | Impact height (m) | Impact velocity (m/s) | Impact kinetic energy (kJ) |
|---|---|---|---|---|
| CP-V10 |  |  | 10 | 130 |
| CP-V15 |  |  | 15 | 292.5 |
| CP-V20 | 3# pile center |  | 20 | 520 |
| CP-V25 |  |  | 25 | 812.5 |
| CP-V30 |  | 5.25 | 30 | 1170 |
| CS-V10 |  |  | 10 | 130 |
| CS-V15 |  |  | 15 | 292.5 |
| CS-V20 | 2# slab center |  | 20 | 520 |
| CS-V25 |  |  | 25 | 812.5 |
| CS-V30 |  |  | 30 | 1170 |

Note: CP denotes the 3# pile center as impact location; CP denotes the 2# slab center as impact location; V denotes
the velocities of rockfall.
*2.2. Model validation*
In order to verify the rationality of the selected material constitutive model and the
established numerical model. Two physical model tests from previously published papers (Heng et
al., 2021; Demartino et al., 2017; Schellenberg, 2008) were selected to reproduce.
*2.2.1.    Failure test of RC cantilever column*
The physical model test conducted by Demartino et al. (2017) was selected to verify the
ability of constitutive model to reflect the accumulative damage for RC structures under impact
loads. The model is composed of a cylindrical column with a diameter of 0.3 m and a height of 1.7
m, and a square-section concrete foundation with length of 0.9 m and height of 0.5 m. The column
was reinforced with sixteen 8 mm diameter longitudinal reinforced bar and 6.5 mm diameter
stirrups at 100 mm spacing. The foundation was firmly connected to the ground using four 50 mm
diameter high-strength prestressed reinforced bar. The experiment involved a test truck made of
Q235 steel (considered as a rigid body) (Fig. 5a). The impactor was positioned 0.4 m above the
bottom of the column and was released at a velocity of 3.02 m/s (impact energy of 7.21 kJ). Fig.
5b shows the numerical model with hexahedral mesh. The material constitutive models for
components are shown in Table 1. For the boundary conditions, the model was fixed with four
high-strength bolts.
The trend and amplitude of the impact forces by numerical simulations closely matched the
experimental results (Fig. 6). Similarly, Table 3 Simulation results of different mesh sizes.

| Items | Impact force (kN) | Displacement of column at 1.2m height (mm) | Number of the element | Computational time (hour) |
|---|---|---|---|---|
| Physical model test | 999.52 | 22.3 | / | / |
| 25 mm mesh size | 966.72 | 23.1 | 5462900 | 24 |
| 50 mm mesh size | 978.1 | 22 | 807534 | 4.2 |
| 100 mm mesh size | 1009.35 | 21.3 | 172268 | 1.2 |

**Table 4** indicates a consistency between the extent of the experimental and numerical
damage in concrete. The deviations of peak impact forces between the numerical simulations and
the experiments were below 10% (Table 3). These results suggest that the numerical model and its
governing parameters can reliably simulate the accumulative damage in RC structures subjected to
impact loads. Considering both accuracy and computational time, a mesh size of 50 mm was
selected for the numerical simulations conducted in this study.

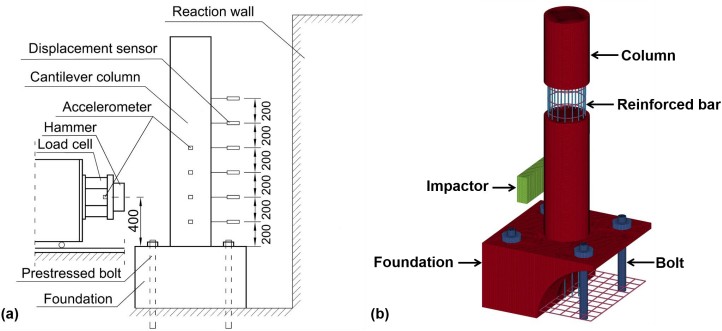

**Fig. 5.** Model of RC cantilever column failure test
(a) experimental model (b) numerical model (unit: mm).

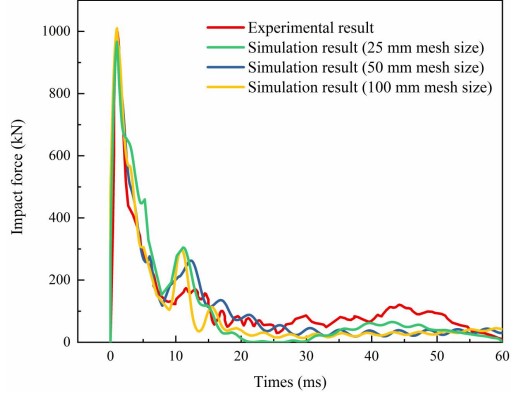

**Fig. 6.** Dynamic curve of impact force with different mesh size.

177            **Table 3** Simulation results of different mesh sizes.

| Items | Impact force (kN) | Displacement of column at 1.2m height (mm) | Number of the element | Computational time (hour) |
|---|---|---|---|---|
| Physical model test | 999.52 | 22.3 | / | / |
| 25 mm mesh size | 966.72 | 23.1 | 5462900 | 24 |
| 50 mm mesh size | 978.1 | 22 | 807534 | 4.2 |
| 100 mm mesh size | 1009.35 | 21.3 | 172268 | 1.2 |

**Table 4** Comparison of experimental and simulation results of concrete damage accumulation with time.

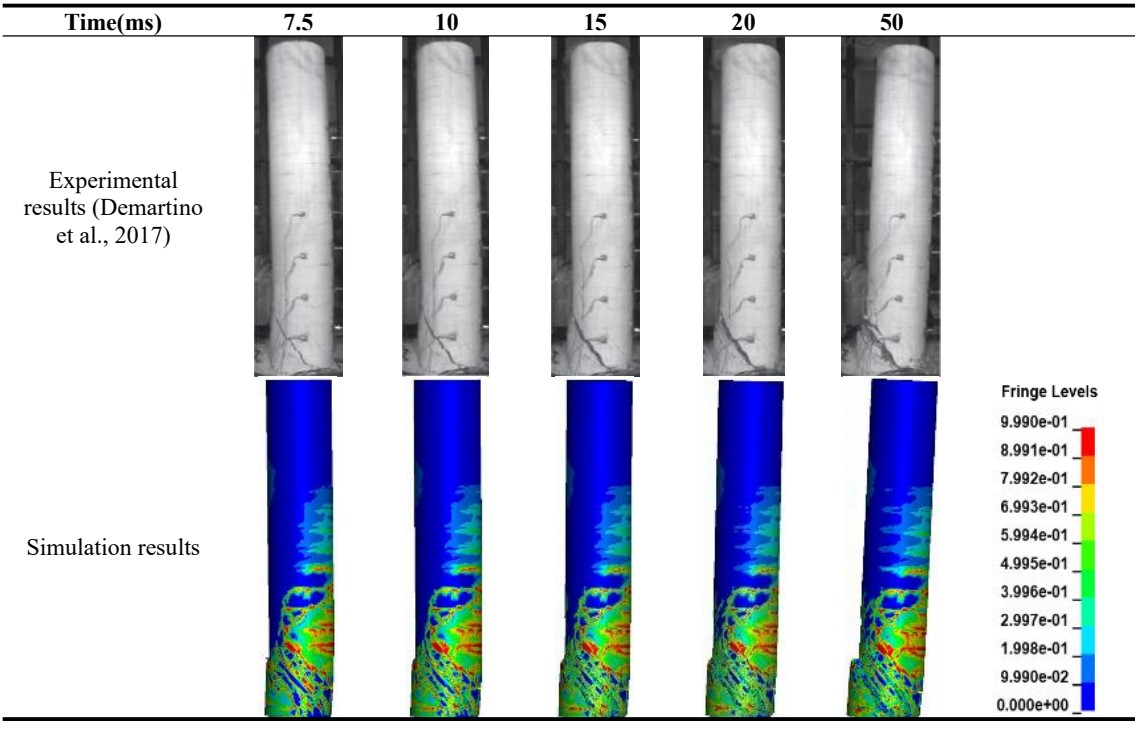

*2.2.2. Failure test of RC slab with a buffer layer*
The physical model test conducted by Schellenberg (2008) was selected to validate the
capability of the constitutive model to reflect the interaction among the boulder, sand buffer layer,
and RC structure. The specimen comprises a RC slab measuring 1.5 m × 1.5 m × 0.23 m and a
sand buffer layer with 0.5 m in radius and 0.45m in thickness (Fig. 7). The slab is reinforced with
one layer of reinforced bar with 12 mm diameter and a spacing of 95 mm for the lower layer. The
diameter and density of the boulder are 0.8 m and 3110 kg/m$^3$, respectively. The impact position is
located at the center of the buffer layer, with an impact velocity of 5.5 m/s (impact energy of 14.4
kJ). The material constitutive models for concrete, reinforced bar, and sand buffer layer are shown
in Table 1. For the boundary conditions, the bottom of the supports was fixed.
Fig. 8 presents the dynamic curve of impact force, displacement of slab center, and axial
strain of center reinforced bar. The results demonstrate that the deviations of the peak impact force,
the maximum strain of reinforced bar, and the slab center displacement are less than 10%.
Therefore, the numerical model and its governing parameters are deemed reliable for simulating
the behavior of a sand cushion layer and an RC structure under impact loads.

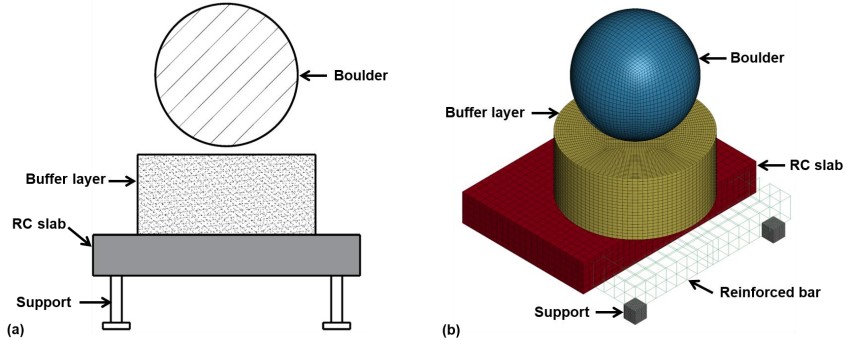

**Fig. 7.** Model of RC slab failure test
(a) experimental model (b) numerical model (unit: mm).

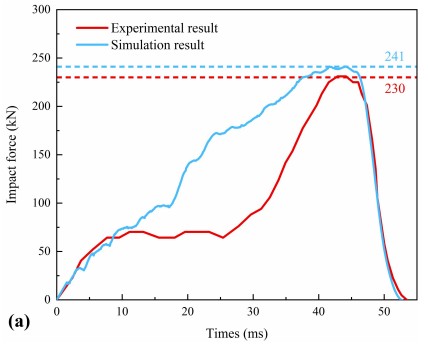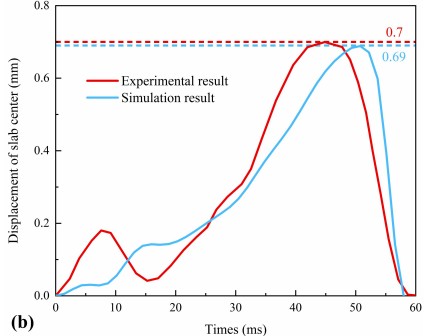

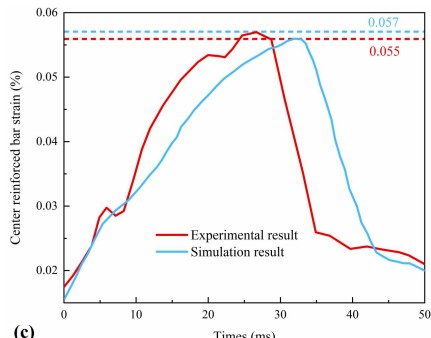

**Fig. 8.** Comparisons between experimental and simulation results
(a) impact force (b) displacement of slab center (c) axial strain of reinforced bar.

## 3.    Numerical results

In this section, the dynamic response of PSRW under different impact centers and different

impact velocities are compared and analyzed. The main evaluation indexes are as follows: impact

force (the contact force between the impactor and the buffer layer), interaction force (the contact

force between the buffer layer and the RC structure), stress of concrete and reinforced bar,

concrete damage, lateral displacement at the crown of different components (piles and slabs), and

lateral displacement of all piles at the ground surface.

*3.1. Influence of different impact centers*

To analyze the influence of dynamic behaviors of PSRW under different impact centers, two

group simulations under maximum impact energy (CP-V30 and CS-V30) are selected for

comparison.

*3.1.1.    Impact force and interaction force*

Fig. 9a and 9b show the dynamic curves of the impact force and interaction force,

respectively. Both force curves exhibit a distinct single-peaked pattern. The impact force rapidly

reduces to zero due to the energy-dissipating properties of the sand buffer layer (Fig. 9a). In

contrast, the interaction force remains at a non-zero value (475 kN) (Fig. 9b). Owing to the

permanent deformation sustained by the structure, and the gravitational force exerted by the sand

buffer acts on the surface of the structure. Furthermore, Fig. 9a illustrates the close overlap of the

impact forces for various impact centers, depending on the buffer and impactor characteristics, and

minimally affected by the impact center. The slight differences observed in the dynamic curve of

interaction force under CP-V30 and CS-V30 may be attributed to the flexural stiffness of the slab

and pile.

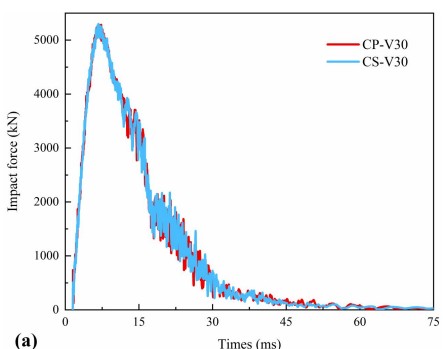
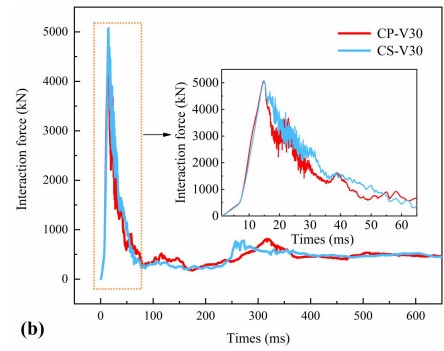

**Fig. 9.** Dynamic curves of impact force and interaction force under various impact centers
(a) impact force (b) interactional force.

### *3.1.2.    Stress of concrete*

The minimum principal stress of concrete and the effective stress of reinforced bar are important indexes to evaluate the dynamic response of RC structures (Zhong et al., 2021; Zhong et al., 2022). Fig. 10 shows the minimum principal stress nephogram of concrete under CP-V30 from 1 to 650 ms. When t = 1 ms (Fig. 10a), the minimum stress focus on the bottom of the piles. When t = 14.7 ms (Fig. 10b), the minimum principal stress of concrete around the impact point increased rapidly to 7.421 MPa. When t= 22.8 ms (Fig. 10c), the concrete elements at the joints of the 3# pile and slabs achieve compressive strength, leading to concrete damage. When t= 650 ms (Fig. 10d), the total volume of damaged elements reaches 0.63 m$^3$, which occupies a proportion of 0.35%.

Fig. 11 shows the minimum principal stress nephogram of concrete under CP-V30 from 1 to 650 ms. When t = 1 ms, the maximum stress focus on the bottom of the piles (Fig. 11a). When t = 14.7 ms, the minimum principal stress around the impact point increased rapidly to 12.117 MPa (Fig. 11b). When t = 22.4 ms, the elements of the concrete at the impact point of the 2# slab achieve ultimate compressive strength, leading to the concrete damage (Fig. 11c). When t = 650 ms, the total volume of damage elements reaches 0.61 m$^3$ (Fig. 11d), which occupies a proportion of 0.34%.

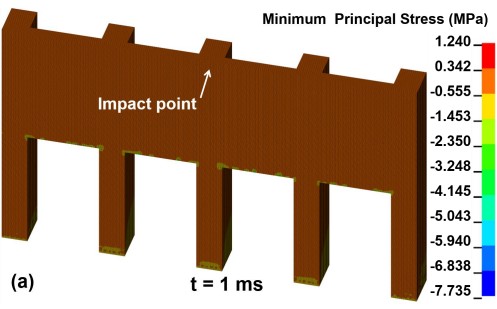
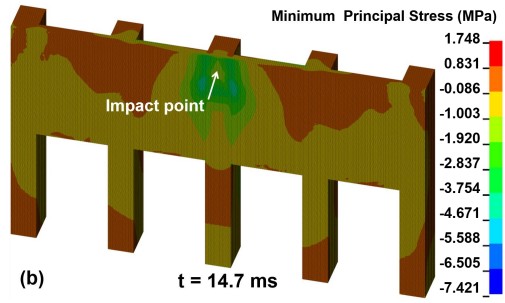

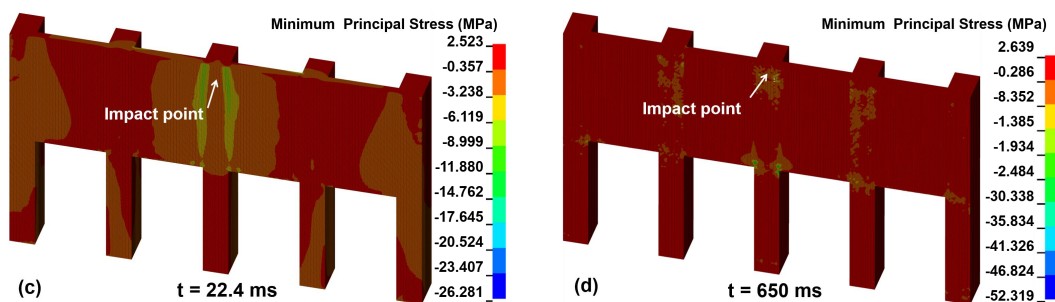

**Fig. 10.** Minimum principal stress nephogram of concrete under CP-V30.


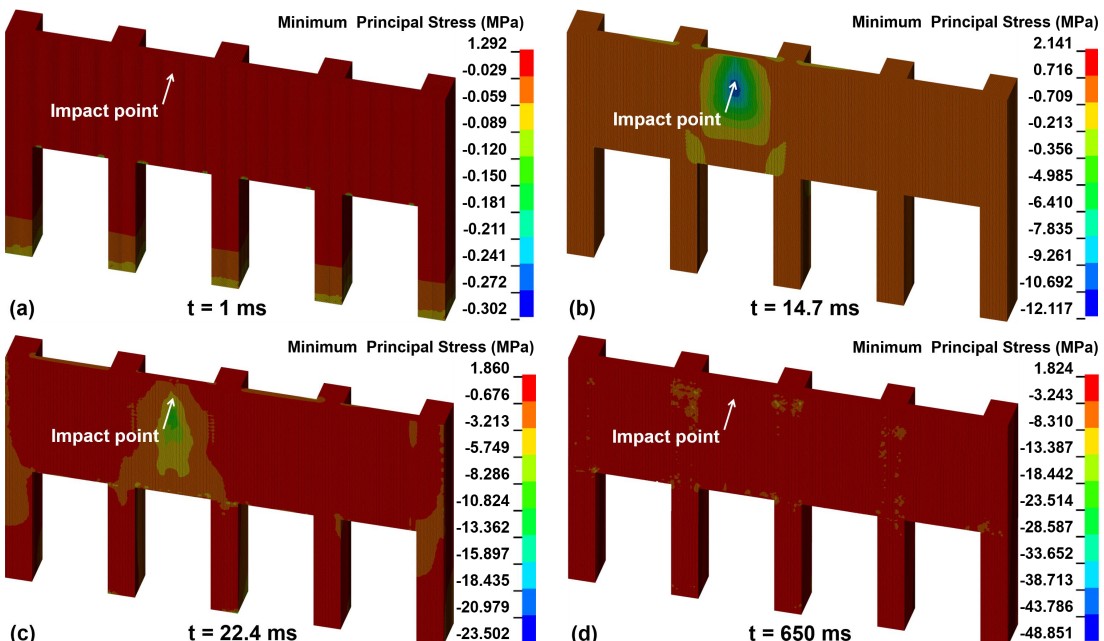

**Fig. 11.** Minimum principal stress nephogram of concrete under CS-V30.

*3.1.3.    Stress of reinforced bar*
Fig. 12 shows the effective stress nephogram of the reinforced bar from 1 to 650 ms under
the condition of CP-V30. It can be observed that: (i) when t = 1 ms, the maximum stress
concentrated at the bottom of the pile (Fig. 12a); (ii) when t = 14.7 ms (the moment of attaining
the maximum interaction force), the maximum stress concentrated at the vicinity of the impact
point and the joints of piles and slabs (Fig. 12c); (iii) when t = 650 ms, the maximum stress
concentrated at the longitudinal bar of 2#, 3#, and 4# pile (Fig. 12d). Noteworthily, the effective
stress of reinforced bar did not exceed the ultimate yield stress.
Fig. 13 shows the effective stress nephogram of reinforced bar from 1 to 650 ms under CS-
V30. It can be observed that: (i) when t = 1 ms, the maximum stress concentrated at the bottom of
the pile (Fig. 13a); (ii) when t = 14.7 ms, the effective stress of reinforced bar around the impact
point increased rapidly to 137.2 MPa (Fig. 13c); (iii) when t = 650 ms, the maximum stress
concentrated at the longitudinal bar of 2#, 3#, and 4# pile (Fig. 13d). Noteworthily, the effective
stress of reinforced bar did not exceed the ultimate yield stress.

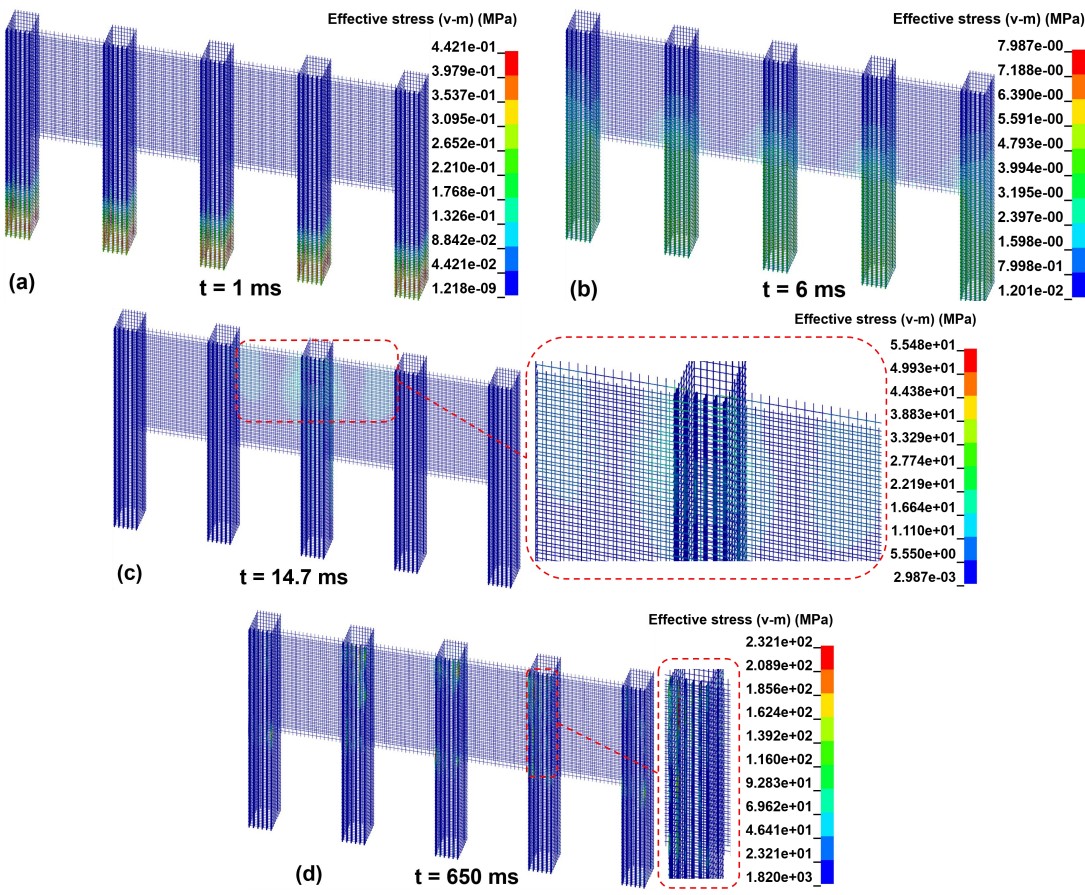

Fig. 12. Effective stress nephogram of reinforced bar under CP-V30.

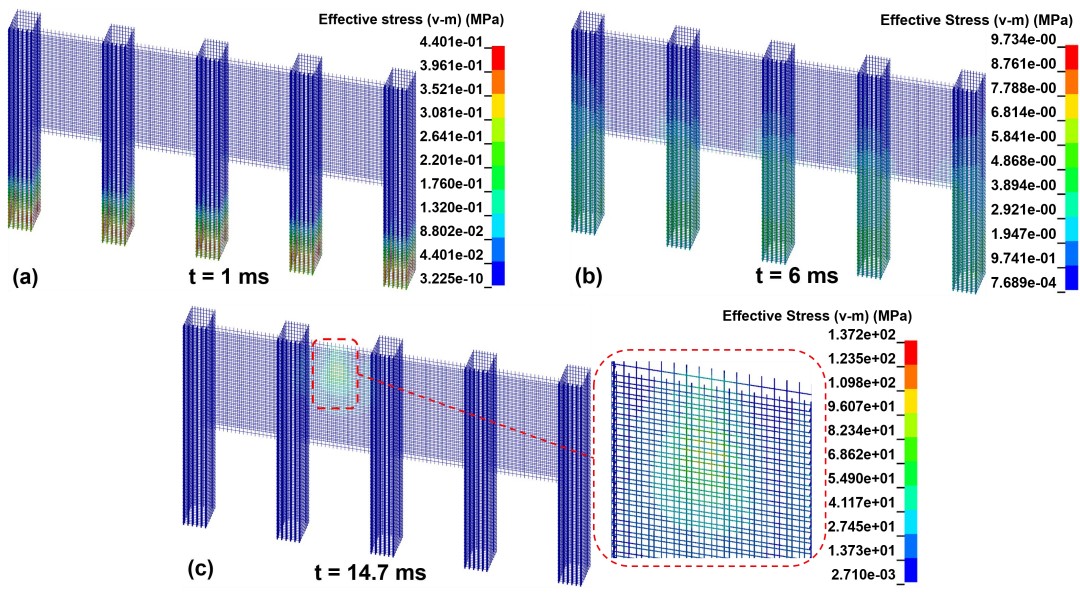

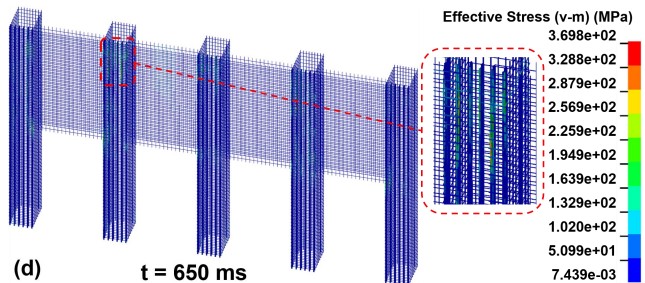

**Fig. 13.** Effective stress nephogram of reinforced bar under CS-V30.

*3.1.4.    Lateral displacement at the crown of different components*

Fig. 14a presents lateral displacements at the crown of different components under CP-V30 and CS-V30 conditions. The lateral displacement rapidly increased till t = 177 ms and gradually decreased until t = 650 ms. The final displacement does not reach 0, indicating plastic deformation of both the pile and the slab. Comparing the lateral displacement under CS-V30 and CP-V30 (Fig. 14), the trends are consistent, but the magnitude differs. This discrepancy in magnitude can be attributed to the greater deformation capacity of slab compared to pile when subjected to the same impact energy.

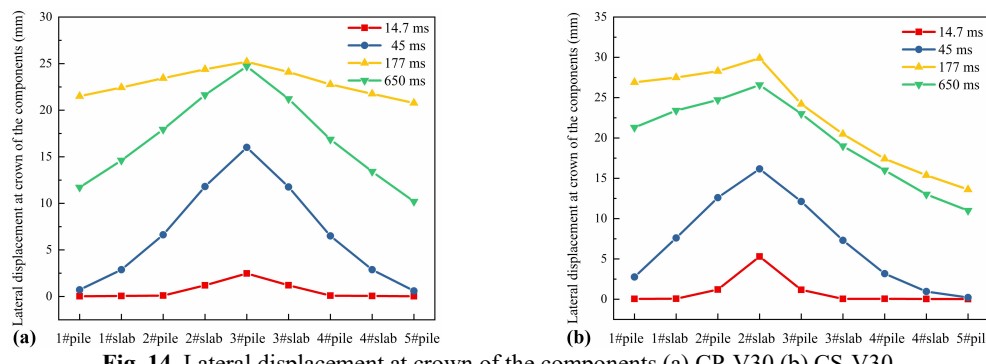

**Fig. 14.** Lateral displacement at crown of the components (a) CP-V30 (b) CS-V30.

*3.1.5.    Lateral displacement of piles at the ground surface*

Fig. 15a and 16b show the dynamic curve of lateral displacement of all piles at the ground surface under CP-V30 and CS-V30, respectively. Under CP-V30, the 3# pile exhibited the maximum lateral displacement, whereas the 2# pile exhibited the maximum lateral displacement under CS-V30. This discrepancy is due to the structural asymmetry on either side of the impact center under CS-V30, which allows one side of pile #2 greater freedom, resulting in larger lateral displacement. When comparing the lateral displacement of 2# pile under CS-V30 and 3# pile under CP-V30 (Fig. 15c), it is apparent that the maximum lateral displacement of pile at the ground surface is greater under CP conditions, despite the same impact velocity. The characteristics of the lateral displacements suggest that the concrete slab is capable of undergoing

larger deformations and absorbing more energy.

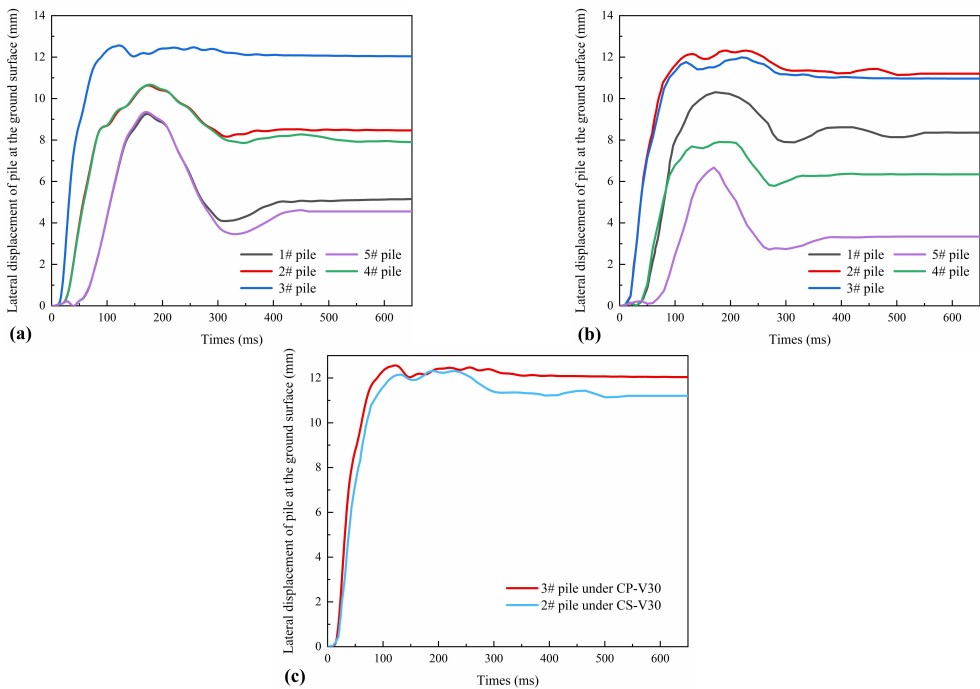

**Fig. 15.** Dynamic curves of lateral displacement of pile at the ground surface
(a) CP-V30 (b) CS-V30 (c) compare between CP-V30 and CS-V30.

*3.2. Influence of different impact velocities*
Figure 17 demonstrates that under CP conditions, the impact force, interaction force, and
lateral displacement of pile #3 at the ground surface increase as the impact velocity of rockfall
rises. When the velocity increases from 15 m/s to 30 m/s, the impact force increases by 1.42, 1.91,
and 2.41 times, the interaction force increases by 1.25, 1.47, and 1.68 times, and the lateral
displacement of 3# pile at ground surface increases by 1.57, 2.24, and 3 times at t = 650 ms. By
comparing the magnitude of changes, the lateral displacement is more sensitive to velocity
variations than impact force and structural interaction force.

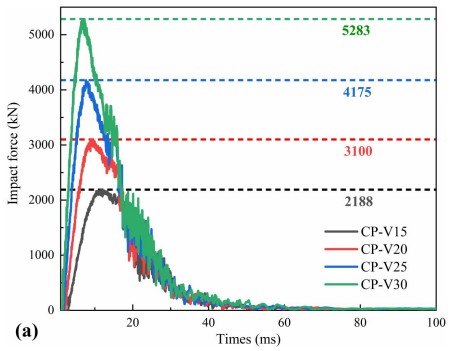
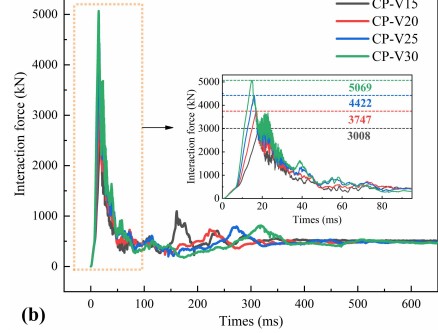

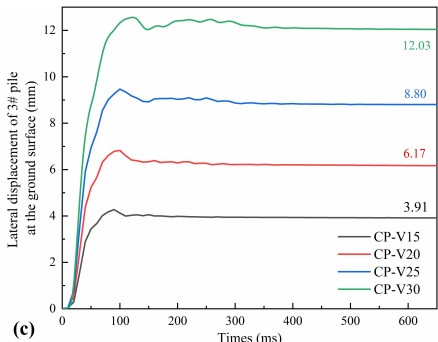

**Fig. 16.** Dynamic curves of evaluation indexes under various velocities
(a) impact force (b) interactional force (c) lateral displacement at the ground surface of 3# pile.

275        Fig. 17 shows the impact force, interaction force, and lateral displacement of 2# pile at the

ground surface enlarge as the impact velocity increases under CS conditions. When the velocity
increases from 15 m/s to 30 m/s, the impact force increases by 1.41, 1.90, and 2.41 times, the
interaction force increases by 1.24, 1.47, and 1.68 times, and the lateral displacement of 3# pile at
ground surface increases by 1.55, 2.23, and 3 times at t = 650 ms. Similar to the CP conditions, the
lateral displacement is still most sensitive to velocity variations.

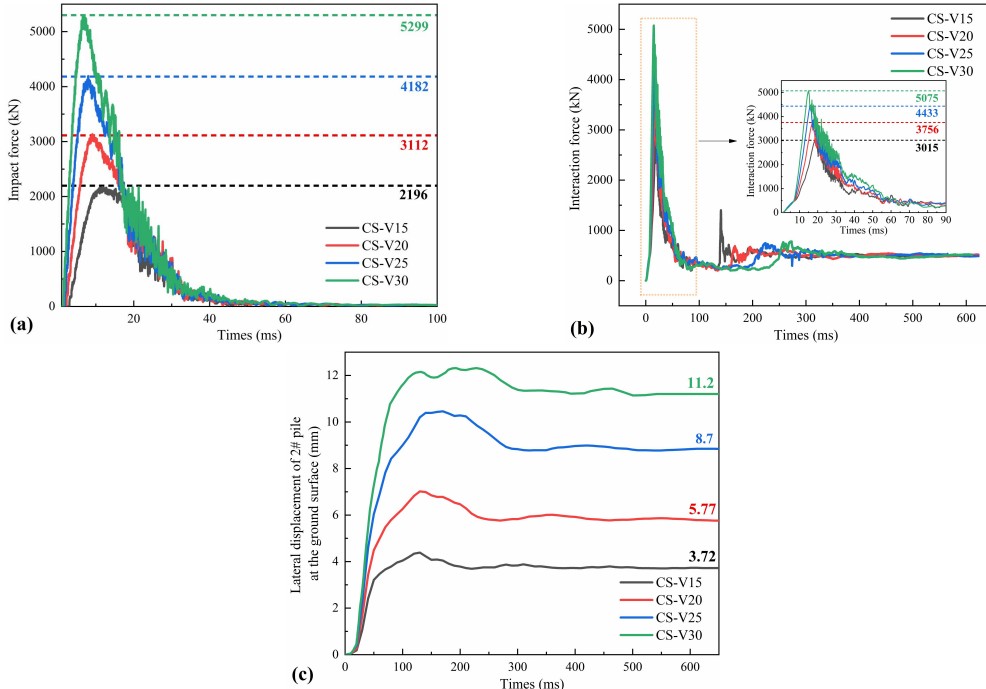

**Fig. 17**. Dynamic curves of evaluation indexes under various velocities
(a) impact force (b) interactional force (c) lateral displacement at the ground surface of 3# pile.

**4.  Discussions**
*4.1. Comparison of impact force calculation models*

283        A comparative analysis compared the elastic theories proposed by Labiouse et al. (1996),

Kawahara and Muro (2006), Pichler et al. (2006), and Hertz (1881) was conducted to assess the
validity of the numerical simulation (Fig. 18). The results reveal a fundamental linear correlation
between impact force and velocity. Overall, the computational results are consistent with those of
other models in terms of magnitude, thus confirming the validity of the calculations reported here.

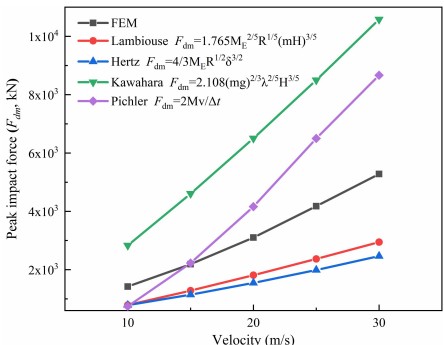

**Fig. 18.** Relationship between impact velocity and impact force.

*4.2. Relationship between structural evaluation indexes and impact energy*
Table 5 lists the initial kinetic energy of impactor ($E$), the peak impact force ($F_{dm}$), the peak
interaction force ($F_{im}$), the ratio of the peak impact force to the peak interaction force ($\alpha$), the
maximum the lateral displacement of pile at the ground surface at t = 650 ms ($S_{mpt}$), the number of
damage failure units ($N_d$), and the ratio of damage failure units to overall RC structure units ($\beta$).
**Table 5** Simulation results of various impact cases.

| Case | $E$ (kJ) | $F_{dm}$ (kN) | $F_{im}$ (kN) | $\alpha$ (%) | $S_{mpt}$ (mm) | $N_d$ | $\beta$ (%) |
|---|---|---|---|---|---|---|---|
| CP-V10 | 130 | 1420 | 2170 | 65.4 | 2.25 | 83 | 0.0059 |
| CP-V15 | 292.5 | 2188 | 3008 | 72.7 | 3.91 | 817 | 0.0577 |
| CP-V20 | 520 | 3100 | 3747 | 82.7 | 6.17 | 2179 | 0.1539 |
| CP-V25 | 812.5 | 4175 | 4422 | 94.4 | 8.8 | 3088 | 0.2181 |
| CP-V30 | 1170 | 5283 | 5069 | 104.2 | 12.03 | 5040 | 0.3559 |
| CS-V10 | 130 | 1426 | 2182 | 65.4 | 1.76 | 52 | 0.0037 |
| CS-V15 | 292.5 | 2196 | 3015 | 72.7 | 3.72 | 321 | 0.0227 |
| CS-V20 | 520 | 3112 | 3756 | 82.7 | 5.77 | 1062 | 0.0750 |
| CS-V25 | 812.5 | 4182 | 4433 | 94.4 | 8.7 | 2728 | 0.1927 |
| CS-V30 | 1170 | 5299 | 5075 | 104.2 | 11.2 | 4880 | 0.3446 |

Under the premise of known impact energy, estimating impact force, interaction force, and
displacement of pile for the structural design is very important. As shown in Table 5, the variation
in peak impact force ($F_{dm}$) with different impact centers is minimal. Consequently, CP simulation
results were chosen for further analysis. The dependence of the peak impact force on the impact
energy is shown in Fig. 19a, with a correlation coefficient $R^2 = 0.99$, i.e.,
$$F_{dm} = 3.69(E + 290.33) = 1845(mv^2 + 0.58) \tag{1}$$

where $m$ is the impactor mass ($m$= 2600 kg herein); $v$ is the initial impact velocity (10 m/s $\leqslant$
$v \leqslant$ 30 m/s herein).
The dependence of the ratio of peak impact force to peak interaction force on the impact
energy is shown in Fig. 19b, with a correlation coefficient of 0.99, i.e.,
$$\alpha = 0.037(E + 1671.89) = 18.5(mv^2 + 3.34) \tag{2}$$

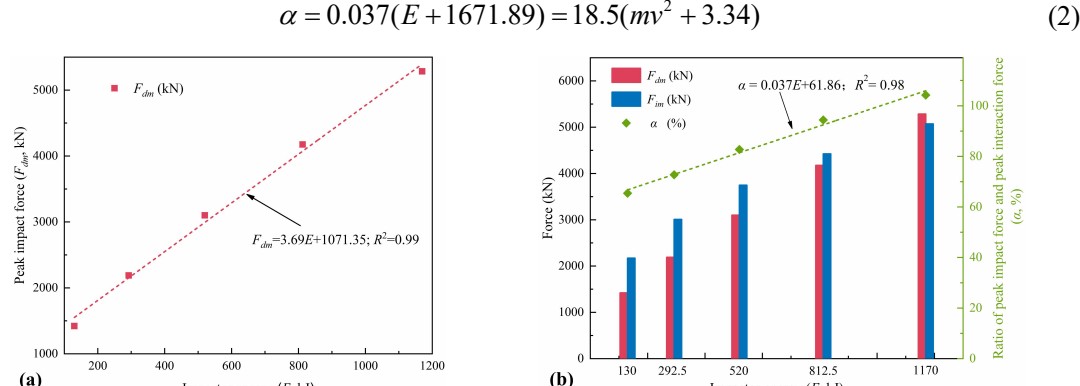

**Fig. 19.** Dependence of various indexes on impactor energy (a) peak impact force (b) the ratio of peak impact force and peak interaction force.

The lateral displacement of pile at the ground surface is an important index to judge the
failure of pile foundation under lateral load. As shown in Table 5, the maximum lateral
displacement of pile at the ground surface under pile as impact center is greater than that under
slab as impact center. Therefore, the situation where the pile is the center of impact is the more
dangerous. As shown in Fig. 20, with the increase of impact energy, the displacement value and
number of damage failure units enlarges, which means the structure suffers more damage under
CP. Furthermore, the maximum lateral displacement of pile at the ground surface when t = 650 ms,
can be calculated by the following equation:
$$S_{mpt} = 0.00934(E + 164.88) = 4.67(mv^2 + 0.33) \tag{3}$$

**Fig. 20.** Dependence of the lateral displacement of 3# pile at the ground surface on impactor energy

According to the Chinese Specification for the Design of Rock Retaining Wall Engineering in
Geological Hazards (CAGHP, 2019), the lateral displacement of the resistant sliding pile at the
ground surface must not exceed 10 mm. Substituting this value into Formula 3, the maximum
impact energy that the PSRW can withstand in this study is 905 kJ.
*4.3. Comparison with other concrete rockfall retaining walls*
Table 6 presents crucial data on an improved cast-in-place rockfall concrete barrier developed
by the US Department of Transportation (Patnaik et al., 2015). This barrier exhibits relatively low
resistance to impact energy, which restricts its applicability to situations where high-impact energy
rockfalls are likely to occur. Integrating a specialized buffering layer on the concrete retaining wall,
the barrier's impact resistance can be effectively enhanced (Kurihashi et al., 2020). According to
Maegawa et al. (2011), concrete rockfall barriers with a buffering layer offer a maximum impact
resistance ranging from approximately 120 to 490 kJ. Addressing the resistance limitations of
traditional concrete rockfall barriers, Furet et al. (2022) proposed the articulated concrete block
rockfall protection structures. These innovative structures allow concrete blocks hingedly
connected to one another, enabling greater impact energy absorption.

**Table 6** Comparison of different concrete rockfall protection structures

| Structure name | The maximum impact energy that structure can withstand (kJ) | Energy dissipation ratio (%) | Interception altitude (m) |
|---|---|---|---|
| Cast-in-place rockfall concrete barriers (Patnaik et al., 2015) | 127 | / | 0.81 |
| Concrete retaining wall with buffering system (Kurihashi et al., 2020) | 273 | 100 | 2.5 |
| Concrete rock – wall (Maegawa et al., 2011) | 490 | / | / |
| Articulated concrete blocks rockfall protection structure (Furet et al., 2022) | 1020 | 100 | 3.2 |
| Pile-slab retaining wall | 905 | 100 | 6 |

Note: Energy dissipation ratio denotes the ratio of dissipated energy to input energy.
In terms of energy dissipation, structure damage and friction are responsible for 74% of the
impact energy dissipation, with the remaining 26% attributed to other phenomena such as
deformation of structural elements, elastic wave propagation, viscous damping, and fracturing.
Compared to conventional concrete rockfall barriers, PSRW exhibit significantly higher impact
resistance (905 kJ) and interception height (6 m). Similarly, these structures absorb all the impact
energy, preventing the impactor from rebounding.
For traditional RC retaining walls subjected to a 16 kJ impact energy, shear cracks develop
diagonally from the impact point, with wider spreading observed on the rear face compared to the
collision surface (Kurihashi et al., 2020). Fig. 21 illustrates the concrete damage nephogram of
PSRW under the impact load of 1170 kN. It is evident that concrete damage primarily
concentrated around the impact point and at the junction between the pile and slab. Importantly,
there is no evidence of crack penetration into the structure itself, indicating that the PSRW
maintains its structural integrity.

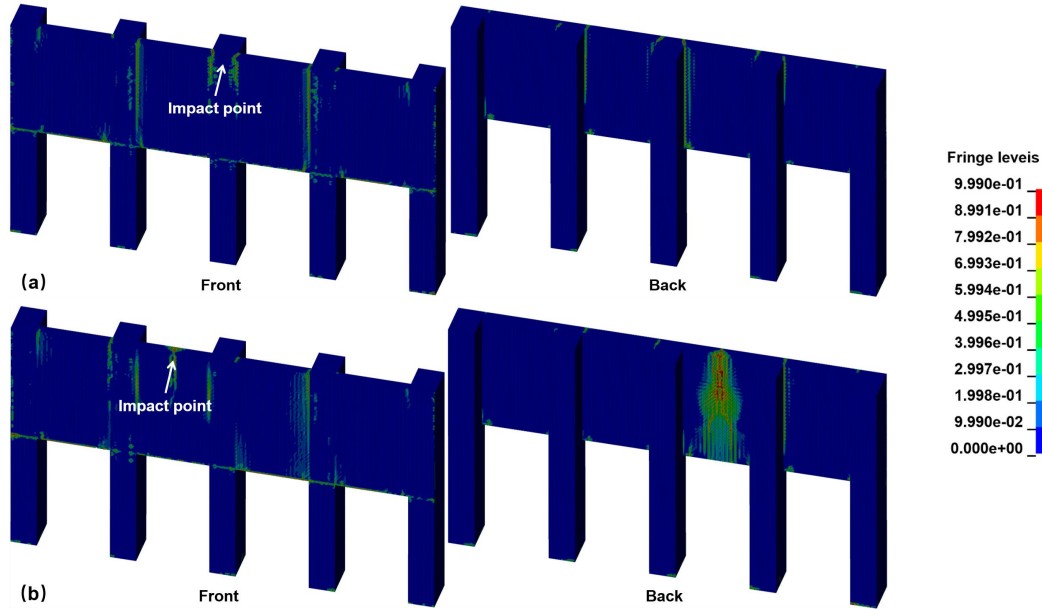

**Fig. 21.** Damage nephogram of concrete at t = 650 ms (a) CP-V30 (b) CS-V30.

Although the lateral displacement of the pile exceeds the stipulated limit, reaching 12 mm as
indicated in Table 5 and Figure 21, it is essential to acknowledge that the specified ultimate lateral
displacement is frequently a conservative estimation. Concurrently, the maximum lateral
displacement at the crown of the cantilever section is 35 mm, which is substantially less than the
lateral displacement threshold for the cantilever section of the anti-slide pile. This threshold is
defined as 1% of the cantilever section's length, according to CAGHP (2019). As a result, the
impact load does not compromise the integrity of the structure.
In summary, the PSRW is an innovative rockfall protection structure, providing an enhanced
level of impact resistance, increased interception height, and reduced concrete damage.
Additionally, the minimal lateral displacement observed after impact further ensures the structural
integrity and safety in challenging terrain areas.
*4.4. Discussion on Engineering Practicality*
The data presented in Table 7 reveal the distribution of rockfall energy levels across four
regions that experience frequent rockfalls. Notably, the Alps region experiences substantial
rockfalls, with many of them exhibiting an impact energy below 1000 kJ. Schneider et al. (2023)

utilized Doppler radar technology to monitor rockfall activity in Brienz/Brinzals, Switzerland. Their findings indicated that although the volume of rockfalls ranged from 1 to 100 $m^3$, smaller events (1 $m^3$) were markedly more common. As previously mentioned, the PSRW can withstand rockfalls with an impact energy of about 1000 kJ, making it an ideal solution for a multitude of small alpine rockfall scenarios. Additionally, its compact size and robust structural stability further enhance its suitability for mountainous construction projects. In cases where the impact energy of falling rocks exceeds 1000 kJ, it is advisable to optimize the mechanical properties of the cushion layer, improve the elastic modulus of concrete, increase the reinforcement ratio of longitudinal tension bars, enlarge the section size of pile at ground level, or add anchoring measures to enhance the bending resistance of the retaining structure.

**Table 7** Rockfall events in different areas

| Study area | Total number of rockfall events | Rockfall energy < 1000 kJ | Percentage |
|---|---|---|---|
| French Alps (Le Roy et al., 2019) | 18 | 9 | 50% |
| Swiss Alps (Dietze et al., 2017) | 37 | 37 | 100% |
| Along the railway in Japan (Muraishi et al., 2005) | 173 | 158 | 91% |
| New South Wales, Australia (Spadari et al., 2013) | 211 | 200 | 94% |

## 5. Conclusion

Compared to existing rockfall protection structures, the PSRW offers enhanced stability and requires a smaller footprint, making it adept at addressing a broad spectrum of rockfall impact scenarios commonly encountered in alpine canyon regions. In this paper, the dynamic response of the PSRW under different impact centers and velocities were compared and analyzed using the FEM simulation method. Additionally, the influencing factors such as peak impact force, peak interaction force, ratio of peak impact force to peak interaction force, concrete stress, reinforcement stress, maximum lateral displacement of the pile at the ground surface, and ratio of damage failure units to overall structure units were quantified. Notably, the formula for calculating the peak impact force of the PSRW (Eqs. 1), the ratio of peak impact force to peak interaction force (Eqs. 2), maximum lateral displacement of the pile at the ground surface (Eqs. 3) based on the impact energy of rockfalls were proposed. The key findings of this study are as follows:

(1) The impact force, interaction force and lateral displacement exhibit a linear correlation with the impact velocity. however, the lateral displacement is more sensitive to velocity variations

than the impact force and interaction force.
(2) Under different impact centers, the variations in impact force and interaction force are
minimal. When the pile serves as the impact center, the lateral displacement of pile at the ground
surface and the extent of concrete damage are significantly greater than when the slab center is the
impact center. This indicates that impacts centered on the pile pose a more hazardous impact
scenario.
(3) Concrete damage predominantly concentrates at the joints between piles and slabs, the
impact center itself, and the section of piles at the ground surface. To minimize structural concrete
damage, it is imperative to prioritize these critical sections in the structural design.
(4) The impact force, the ratio of peak impact force to peak interaction force, and the
maximum lateral displacement of the pile at the ground surface have a significant correlation with
the impact energy. These relationships are crucial for evaluating impact force, interaction force,
and the lateral displacement of piles at ground surface during the design of PRSW structures.
According to Chinese specifications for displacement requirements, the maximum lateral
displacement of the pile at the ground surface should not exceed 10 mm. Consequently, the
maximum impact energy that the PSRW can withstand is 905 kJ, when the crown is designated as
the impact center.

## CRediT authorship contribution statement

**Peng Zou:** Methodology, Simulation, Visualization, Writing - original draft. **Gang Luo:**
Tests design, funding acquisition, writing - review. **Yuzhang Bi:** Visualization, Writing - review.
**Hanhua Xu:** Writing - review.

## Declaration of Competing Interest

The authors declare that they have no known competing financial interests or personal
relationships that could have appeared to influence the work reported in this paper.

## Acknowledgments

This research was funded by the National Natural Science Foundation of China (42277143),
the National Key R&D Program of China (2022YFC3005704), the Sichuan Province Science and
Technology Support Program (2024NSFSC0100) and the Science and the research project of the
Department of Natural Resources of Sichuan Province (KJ-2023-004, KJ-2023-029). The authors
also thank the editors and anonymous reviewers for their constructive comments that improved the
manuscript.

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
