# Peer review of "Dynamic Response of Pile-Slab Retaining Wall Structure"

_EGUsphere, 2023_

## Referee Comment (RC1)

**Review of the manuscript ID egusphere-2023-2715 '*Dynamic Response of Pile-Slab Retaining Wall Structure under Rockfall Impact*' submitted to *Natural Hazards and Earth System Sciences*.**
* * *
Recommendation: ACCEPT

Focus of the paper: numerical modelling in engineering geology.

Relevance: The presented study is the original primary research within scope of the journal.

Title: the title and abstract of this paper clearly reflect its content.

Abstract is well written and clearly describes the undertaken study: In the presented experiment, the impact resistance of the structure is optimized compared to traditional reinforced concrete retaining walls.

Structure: The article is well organized with structured sections. The structure of the manuscript conforms to the journal standards and discipline norm.

Introduction presents a background, defines research goals and provides a clear statement of research problem. It describes the purpose of the research investigation supported by literature. The Introduction well describes the research. Introduction and background show context of the article. Literature is well referenced and relevant.

Research questions and goal are identified. Objectives are relevant to the study aim.

Literature regarding the relevant topics is reviewed, formatted according to the journal rules and appropriately referenced. Major sources include published papers on geotechnical engineering.

Research gaps and weakness in former works are described: the authors evaluated the that the impact position which has a significant effect on the stability of the structure, which has not been investigated earlier.

Motivation is explained: The authors presented numerical experiments to investigate the dynamic response of a pile-slab retaining wall under the impact of rockfall.

Methods: The authors performed a full-scale numerical model of a four-span pile-slab retaining wall satisfying specification requirements.

Results are reported: The authors reported that during the impact process, the stress, strain, and concrete damage of the structure spread from the impact centre to the entire structure and result in permanent deformation. The authors also reported that lateral displacement of pile at ground surface and the number of damage failure units under the pile as the impact centre is greater than those under the slab as impact centre.

Discussion interpreted the major outcomes of this study: The authors discussed a presented series of numerical experiments to investigate the dynamic response of a pile-slab retaining wall under different impact centers and velocities.

Conclusion The authors predicted the maximum impact energy that the structure can resist.

Actuality: the authors found that the impact force, interaction force, lateral displacement of pile at ground surface, and concrete damage is increased with the increase of impact velocity.

Novelty: The authors investigated the relationship between the impact velocity and the maximum lateral displacement of pile at ground surface. The authors also estimated maximum impact energy that the pile-slab retaining wall can withstand.

Academic contribution: Rigorous investigation performed to a high technical and professional standard. The paper deserved to be published in *Natural Hazards and Earth System Sciences*.

Figures The authors presented 20 figures which are of acceptable quality, easy to read, relevant and suitable. Figures are labelled and appropriately described.

Recommendation: This manuscript can be ACCEPTED based on the detailed report above.

With kind regards,

- Reviewer.

13.12.2023.

---

## Author Comment (AC2)

Dear Reviewer:

We gratefully thank for your constructive remarks and useful suggestions, which has significantly raised the quality of the manuscript and has enable us to improve the manuscript. Below the comments of the reviewers are response point by point and the revisions are indicated.

**Reviewer 1**

**1. General Comments:**

I suggest rewriting the abstract section. The current abstract describes too much experimental process, information, and results. These pieces of information are not what readers most want to see, nor are they the most valuable conclusions drawn in this manuscript. Therefore, I suggest the author rewrite the abstract.

**1. Reply:**

We gratefully appreciate for your valuable suggestion. A rewritten abstract is as follows: The numerical experiments investigate the dynamic response of a pile-slab retaining wall under the various impact conditions of rockfall. Results reveal that: (1) during the impact process, the stress, strain, and concrete damage of the structure gradually spread from the impact center to entire structure and ultimately result in permanent deformation; (2) the lateral displacement of pile at the ground surface and the concrete damage under the pile as the impact center is greater than those under the slab as impact center, implying that the impact location has a significant influence on the stability of the structure; (3) there is a positive correlation between the response indexes (impact force, interaction force, lateral deformation of pile and slab, concrete damage, and the impact velocities; (4) within the discussed impact scenarios, the rockfall peak impact force, the ratio of peak impact force to peak interaction force, and lateral displacement of pile at the ground surface had strong linear relationships with rockfall energy. Utilizing this relationship, the estimated maximum impact energy that the pile-slab retaining wall can withstand is 905 kJ in this study when the structure top is taken as the impact point.

**2. General Comments:**

The manuscript lacks some case studies information and is detached from the case and reality. The numerical simulation results of this manuscript appear to lack basis, greatly reducing their value, and the reliability of the results cannot be verified from real projects.

**2. Reply:** Thank you for your comment. As illustrated in Fig. 1, pile-slab retaining wall have gained widespread application in various areas. However, challenges persist in their design and construction, notably regarding the maximum impact energy tolerance of structure and the vulnerability of specific components to damage under impact loads. Therefore, comprehensive research is essential to address these issues. This study is based on the modeling of norms and real cases, aiming to reflect the dynamic response characteristics of the structure in the set impact scenario, and provide a basis for the future design, implementation and improvement of the structure.

[Figure]

(a)            (b)

(c)            (d)

[Figure]

[Figure]

**(e)**                                            **(f)**

**Fig. 1 pile-slab rockfall retaining wall has been implemented**

**3. General Comments:**

The results obtained from numerical simulation lack in-depth mechanism analysis and in-depth refinement of understanding. The knowledge obtained from the current results and conclusions is similar to that of common sense, and there is no need to carry out this work, as readers can also recognize.

**3. Reply:** Thank you for your comment. Currently, numerous measures are available for mitigating rockfall disasters, and adapting different forms of protection structures to suit specific engineering contexts is a pivotal challenge. Rockfall impact energy serves as a crucial parameter in this regard. This manuscript determines the impact energy of pile-slab rockfall retaining wall, offering a valuable reference for selecting appropriate rockfall protection structures in the future. Additionally, we have identified a range of structural characteristics under impact loads, providing essential insights for the future design, enhancement, and implementation of such protective structures. Hence, we believe this study holds significant importance.

**4. General Comments:**

The discussion section of this manuscript is relatively weak. It is recommended that the author, based on reading and referring to a large number of literature, describe the advantages and limitations of the data, models, methods, results, etc. involved in this manuscript.

**4. Reply:** We gratefully appreciate for your valuable suggestion. A rewritten discussion is as follows:

**4. Discussions**

Table 1 lists the initial kinetic energy of impactor ($E$), the peak impact force ($F_{dm}$), the peak interaction force ($F_{im}$), the ratio of the peak impact force to the peak interaction force ($\alpha$), the maximum the lateral displacement of pile at the ground surface at t = 650 ms ($S_{mpt}$), the number of damage failure units ($N_d$), and the ratio of damage failure units to overall RC structure units ($\beta$).

**Table 1** Simulation results for various impact cases.

| Case | $E$ (kJ) | $F_{dm}$ (kN) | $F_{im}$ (kN) | $\alpha$ (%) | $S_{mpt}$ (mm) | $N_d$ | $\beta$ (%) |
|------|------|------|------|------|------|------|------|
| CP-V10 | 130 | 1420 | 2170 | 65.4 | 2.25 | 83 | 0.0059 |
| CP-V15 | 292.5 | 2188 | 3008 | 72.7 | 3.91 | 817 | 0.0577 |
| CP-V20 | 520 | 3100 | 3747 | 82.7 | 6.17 | 2179 | 0.1539 |
| CP-V25 | 812.5 | 4175 | 4422 | 94.4 | 8.8 | 3088 | 0.2181 |
| CP-V30 | 1170 | 5283 | 5069 | 104.2 | 12.03 | 5040 | 0.3559 |
| CS-V10 | 130 | 1426 | 2182 | 65.4 | 1.76 | 52 | 0.0037 |
| CS-V15 | 292.5 | 2196 | 3015 | 72.7 | 3.72 | 321 | 0.0227 |
| CS-V20 | 520 | 3112 | 3756 | 82.7 | 5.77 | 1062 | 0.0750 |
| CS-V25 | 812.5 | 4182 | 4433 | 94.4 | 8.7 | 2728 | 0.1927 |
| CS-V30 | 1170 | 5299 | 5075 | 104.2 | 11.2 | 4880 | 0.3446 |

**4.1. Comparison of impact force calculation models**

A comparative analysis was performed to evaluate the validity of the calculations in this manuscript, comparing the elastic theories proposed by (Labiouse et al., 1996), Kawahara and Muro (2006), Pichler et al. (2006), and Hertz (1997)with the computational results obtained in this study (Fig. 1). The findings reveal a fundamental linear correlation between impact force and velocity. In general, the computational results in this manuscript align with those of other models of similar magnitude, thereby validating the calculations presented herein.

[Figure]

**Fig. 1.** Relationship between impact velocity and impact force

**4.2. Relationship between structural evaluation indexes and impact energy**

Under the premise of known impact energy, estimating impact force, interaction force, and displacement for the structural design is very important. As shown in Table 1, the difference of peak impact force ($F_{dm}$) with different impact centers is minimal, so that CP simulation results were selected to analyze. The dependence of the peak impact force on the impact energy is shown in Fig. 2a, with a correlation coefficient $R^2 = 0.99$, i.e.,

$$F_{dm} = 3.69(E + 290.33) = 1845(mv^2 + 0.58) \tag{1}$$

where $E$ is the initial kinetic energy of impactor, kJ; $m$ is the impactor mass, kg; t ($m = 2.6$ therein), $v$ is the initial impact velocity, m/s (10 m/s $\leq v \leq$ 30 m/s herein).

The dependence of the ratio of peak impact force to peak interaction force on the impact energy is shown in Fig. 2b, with a correlation coefficient of 0.99, i.e.,

$$\alpha = 0.037(E + 1671.89) = 18.5(mv^2 + 3.34) \tag{2}$$

[Figure]

**Fig. 2.** Dependence of various indexes on impactor energy (a) peak impact force (b) the ratio of peak impact force and peak interaction force.

The lateral displacement of pile at the ground surface is an important index to judge the failure of pile foundation under lateral load. As shown in Table 5, the maximum lateral displacement of pile at the ground surface under pile as impact center is greater than that under slab as impact center. Therefore, the situation where the pile is the center of impact is the more dangerous. As shown in Fig. 3, with the increase of impact energy, the displacement value and number of damage failure units enlarges, which means the structure suffers more damage under CP. Furthermore, the maximum lateral displacement of pile at the ground surface when t = 650 ms, can be calculated by the following aquation:

$$S_{mpt} = 0.00934(E + 164.88) = 4.67(mv^2 + 0.33) \tag{3}$$

[Figure]

**Fig. 3.** Dependence of the lateral displacement of 3# pile at the ground surface on impactor energy

According to the Chinese standard Code for the Design of Rock Retaining Wall Engineering in Geological Hazards (Caghp, 2019), the lateral displacement of the resistant sliding pile at the ground surface must not exceed 10 mm. Substituting this value into Formula 3, the maximum impact energy that the PSRW can withstand in this study is 905 kJ.

**4.3. Comparison with other concrete rockfall retaining walls**

Table 2 illustrates the improved cast-in-place rockfall concrete barrier by the US Department of Transportation demonstrates relatively low maximum impact energy resistance, limiting its suitability to scenarios with large impact energies (Patnaik et al., 2015). The concrete retaining wall with buffering systems, integrating a specialized buffering layer on the traditional retaining wall, effectively enhances the barrier's

impact resistance (Kurihashi et al., 2020). Moreover, the energy dissipation ratio indicates that the structural system absorbs all input energy. According to Maegawa et al. (2011), concrete rockfall barriers typically offer a maximum impact resistance ranging from approximately 120 to 490 kJ. In response to the limitations of traditional concrete rockfall barriers in withstanding impact, Furet et al. (2022) proposed the articulated concrete block rockfall protection structures, wherein concrete blocks are interconnected with hinges to enable the structure to absorb higher impact energy as a whole. Regarding energy dissipation, structure damage and friction account for 74% of the impact energy dissipation, while the remaining 26% is presumed to be propagated or dissipated through phenomena such as deformation of structural elements, propagation by elastic waves, dissipation by viscosity under quiet conditions, and fracturing. Compared to the aforementioned concrete rockfall protection structures, PSRW offer significantly higher impact resistance (905 kJ) and interception height (6 m). Similarly, the structure absorbs all impact energy, and the impactor does not rebound.

In the case of concrete deformation damage, Cast-in-place rockfall concrete barriers remain structurally intact under the maximum impact energy, as indicated by the failure model, wherein the concrete retains its integrity despite extensive cracking. Conversely, in articulated concrete blocks rockfall protection structures, concrete damage predominantly occurs locally near the impact point. Notably, under a 520 kJ impact energy, the structure recedes approximately 1.2 m in the direction of impact. For traditional RC retaining walls subjected to a 16 kJ impact energy, cracks develop diagonally upward and downward from the impact point, with wider spreading observed on the back than on the collision surface (Kurihashi et al., 2020). These spreading cracks are interpreted as shear cracks penetrating from the collision plane. 错误!未找到引用源。 illustrates the concrete damage nephogram of PSRW under the impact load of 1170 k. It is evident that concrete damage primarily occurs near the impact point and at the joint of the pile and plate, with no penetration of cracks into the structure. Although the lateral displacement of the pile exceeds the limit, reaching

12mm as indicated in Table 5 and Figure 21, it is essential to note that the ultimate lateral displacement specified in the code is often a conservative estimate. At the same time, the maximum lateral displacement at the crown of the cantilever section is 35mm, which falls significantly below the lateral displacement limit of the cantilever section of the anti-slide pile (set at 1% of the length of the cantilever section) (Caghp, 2019). Consequently, the structure remains unaffected by the impact load.

In summary, PSRW represents a novel rockfall protection structure, offers a higher impact protection grade, greater interception height, and reduced concrete damage. Furthermore, the minimal lateral displacement post-impact ensures structural safety in terrain area.

**Table 2** Comparison of different concrete rockfall protection structures

| Structure name | The maximum impact energy that structure can withstand (kJ) | Energy dissipation ratio (%) | Interception altitude (m) |
|---|---|---|---|
| Cast-in-place rockfall concrete barriers (Patnaik et al., 2015) | 127 | / | 0.81 |
| Concrete retaining wall with buffering system (Kurihashi et al., 2020) | 273 | 100 | 2.5 |
| Concrete rock – wall (Maegawa et al., 2011) | 490 | / | / |
| Articulated concrete blocks rockfall protection structure (Furet et al., 2022) | 1020 | 100 | 3.2 |
| Pile-slab retaining wall | 905 | 100 | 6 |

**Note: Energy dissipation ratio denotes the ratio of dissipated energy to input energy.**

**Reference**

CAGHP: Code for design of rock retaining wall engineering in geological hazards (T/CAGHP060-2019), China University of Geosciences Press, Wuhan2019. (in Chinese)

Furet, A., Villard, P., Jarrin, J.-P., and Lambert, S.: Experimental and numerical impact responses of an innovative rockfall protection structure made of articulated concrete blocks, Rock Mechanics and Rock Engineering, 55, 5983-6000, https://doi.org/10.1007/s00603-022-02957-x, 2022.

Kawahara, S. and Muro, T.: Effects of dry density and thickness of sandy soil on impact response due to rockfall, Journal of terramechanics, 43, 329-340, https://doi.org/10.1016/j.jterra.2005.05.009, 2006.

Kurihashi, Y., Oyama, R., Komuro, M., Murata, Y., and Watanabe, S.: Experimental study on buffering

system for concrete retaining walls using geocell filled with single-grain crushed stone, International Journal of Civil Engineering, 18, 1097-1111, https://doi.org/10.1007/s40999-020-00520-9, 2020.

Labiouse, V., Descoeudres, F., and Montani, S.: Experimental study of rock sheds impacted by rock blocks, Structural Engineering International, 6, 171-176, https://doi.org/10.2749/101686696780495536, 1996.

Maegawa, K., Yokota, T., and Van, P. T.: Experiments on rockfall protection embankments with geogrids and cushions, GEOMATE Journal, 1, 19-24, 2011.

Patnaik, A., Musa, A., Marchetty, S., and Liang, R.: Full-scale testing and performance evaluation of rockfall concrete barriers, Transportation research record, 2522, 27-36, https://doi.org/10.3141/2522-03, 2015.

Pichler, B., Hellmich, C., Mang, H. A., and Eberhardsteiner, J.: Loading of a gravel-buried steel pipe subjected to rockfall, Journal of Geotechnical and Geoenvironmental Engineering, 132, 1465-1473, https://doi.org/10.1061/(ASCE)1090-0241(2006)132:11(1465), 2006.

---

## Author Response (AR1)

Dear editor and reviewer:

We gratefully thank for your constructive remarks and useful suggestions, which has significantly raised the quality of the manuscript and facilitated its improvement. Below the comments of the reviewers are responses point by point and the revisions are indicated.

**Reviewer**

**1. General Comments:**

I suggest rewriting the abstract section. The current abstract describes too much experimental process, information, and results. These pieces of information are not what readers most want to see, nor are they the most valuable conclusions drawn in this manuscript. Therefore, I suggest the author rewrite the abstract.

**1. Reply:**

We gratefully appreciate for your valuable suggestion. A rewritten abstract is as follows:

**Abstract:** The pile-slab retaining wall has gained widespread utilization in rockfall mitigation engineering, attributed to its excellent impact resistance, substantial interception height, and reliable structural durability. The numerical experiments investigate the dynamic response of a pile-slab retaining wall under the various impact conditions of rockfall. Results reveal that: (1) during the impact process, the stress, strain, and concrete damage of the structure gradually spread from the impact center to entire structure and ultimately result in permanent deformation; (2) the lateral displacement of the pile at the ground surface and the concrete damage under the pile as the impact center is greater than those under the slab as the impact center, implying that the impact location has a significant influence on the stability of the structure; (3) there is a positive correlation between the response indexes (impact force, interaction force, lateral deformation of pile and slab, concrete damage, and the impact velocities; (4) within the discussed impact scenarios, the rockfall peak impact force, the ratio of peak impact force to peak interaction force, and lateral displacement of pile at the ground surface had strong linear relationships with rockfall energy. Utilizing this relationship, the estimated maximum impact energy that the pile-slab retaining wall can withstand is 905 kJ in this study when the structure top is taken as the impact point.

**2.  General Comments:**

The manuscript lacks some case studies information and is detached from the case and reality. The numerical simulation results of this manuscript appear to lack basis, greatly reducing their value, and the reliability of the results cannot be verified from real projects.

**2.  Reply:**

Thank you for your comment. As illustrated in Fig. 1, pile-slab retaining wall have gained widespread application in various areas. However, there are still challenges in their design and construction, notably regarding the maximum impact energy tolerance of structure and the vulnerability of specific components to damage under impact loads. Therefore, comprehensive research is essential to address these issues. This study utilized a combination of normative modeling and real-world case studies, aiming to capturing the dynamic response characteristics of the structure under specified impact scenarios. The findings serve as a foundation for the future design, implementation, and enhancement of the structural framework.

[Figure]

**(a) 10.8 m height PSRW at Kangding county**

[Figure]

**(b) 4 m height PSRW with tyre cushion at Kangding county**

[Figure]

**(c) 4m height PSRW with stone cushion at Kangding county**

[Figure]

**(d) 5m height PSRW with sand cushion at Kangding county**

[Figure]

[Figure]

**(e) 9m height PSRW with stone cushion at Zhangmu port in Tibet**

**(f) 9m height broken PSRW with no cushion at Zhangmu port in Tibet**

**Fig. 1 pile-slab rockfall retaining wall has been implemented**

**3. General Comments:**

The results obtained from numerical simulation lack in-depth mechanism analysis and in-depth refinement of understanding. The knowledge obtained from the current results and conclusions is similar to that of common sense, and there is no need to carry out this work, as readers can also recognize.

**4. Reply:**

Thank you for your comment. Currently, numerous measures are available for mitigating rockfall disasters. However, adopting different protection structures to suit specific engineering contexts is a pivotal challenge. Rockfall impact energy serves as a key parameter in this regard. This manuscript determines the impact energy of pile-slab rockfall retaining wall, offering a valuable reference for the selection of appropriate rockfall protection structures in the future. Additionally, we have identified a range of structural characteristics under impact loads, offering crucial insights that will inform the future design, enhancement, and implementation of such protective structures. Consequently, we firmly believe that this study possesses considerable significance.

**5. General Comments:**

The discussion section of this manuscript is relatively weak. It is recommended that the author, based on reading and referring to a large number of literature, describe the advantages and limitations of the data, models, methods, results, etc. involved in this manuscript.

**6. Reply:**

We gratefully appreciate for your valuable suggestion. A rewritten discussion is as follows:

[revised manuscript text omitted]

**Editor**

**1.    General Comments:**

In addition, more generalisation would be welcome, namely on how this research would apply in other geographic areas with similar hazards, namely the Alps where rockfall is probable.

**1.    Reply:**

We gratefully appreciate for your valuable suggestion. We have added a practical discussion of the structure to the discussion section. The specific contents are as follows.

*4.5. Discussion on Engineering Practicality*

The statistical table presented in Table 6 illustrates the rockfall energy levels across four regions globally prone to frequent rockfall disasters. Evidently, from Table 6, it is apparent that a significant proportion of rockfall incidents consist of small alpine rockfalls possessing an impact energy below 1000 kJ. Schneider et al. (2023) employed Doppler radar to observe rockfall occurrences within the active rockfall complex situated in Brienz/Brinzals, Switzerland. Their findings revealed that while the rockfall events encompassed volumes spanning from 1 to 100 $m^3$, smaller events (1 $m^3$) proved to be significantly more prevalent. As aforementioned, the PSRW exhibits resilience against rockfalls exerting an impact energy of approximately 1000kJ, thereby rendering it an apt choice for numerous small alpine rockfall situations. Furthermore, its compact size and robust structural stability bolster its suitability for mountainous construction endeavors.

**Table 1** Survey results of rockfall events in different areas

| Study area | Total number of rockfall evens | Rockfall energy < 1000 kJ | Percentage |
|---|---|---|---|
| French Alps (Le Roy et al., 2019) | 18 | 9 | 50% |
| Swiss Alps (Dietze et al., 2017) | 37 | 37 | 100% |
| Along the railway in Japan (Muraishi et al., 2005) | 173 | 158 | 91% |
| New South Wales, Australia (Spadari et al., 2013) | 211 | 200 | 94% |

---

## Referee Report (RR1)

**Review of the manuscript No. egusphere-2023-2715 '*Dynamic Response of Pile-Slab Retaining Wall Structure under Rockfall Impact*' submitted to *NHESS*.**
* * *
Recommendation: accept.

Focus of the paper: The pile-slab retaining wall which is useful for in rockfall mitigation engineering.

Relevance: The presented study is the original primary research within the scope of the journal.

Abstract is well written and clearly describes the undertaken study.

Structure: The article is well organized with structured sections.

Introduction presents a background, defines research goals and provides a clear statement of research problem. It presents the purpose of the research investigation which is supported by the pertinent literature. Literature is well referenced and relevant.

Research questions and goal are identified. Objectives are relevant to the study aim.

Research gaps and weakness in former works are described; the existing gaps are identified.

Motivation: The pile-slab retaining wall is attributed to its excellent impact resistance, substantial interception height, and reliable structural durability.

English language: acceptable. Clear, unambiguous, professional English language used throughout.

Data used in this study are described. Data are explained, sources are mentioned.

Methods: The authors presented numerical experiments which investigate the dynamic response of a pile-slab retaining wall under the various impact conditions of rockfall. Methods are explained.

Results are reported: Results reveal that during the impact process, the stress, strain, and concrete damage of the structure gradually spread from the impact center to entire structure and ultimately result in permanent deformation. Utilizing found relationship, the authors estimated maximum impact energy that the pile-slab retaining wall can withstand

Discussion interpreted the major outcomes of this study: The authors found that the lateral displacement of the pile at the ground surface and the concrete damage under the pile as the impact center is greater than those under the slab as the impact center, implying that the impact location has a significant influence on the stability of the structure.

Conclusion The authors concluded that there is a positive correlation between the response indexes (impact force, interaction force, lateral deformation of pile and slab, concrete damage, and the impact velocities. Conclusions are well stated and linked to original research question.

Actuality, novelty and importance of the research is clear: the authors revealed that within the discussed impact scenarios, the rockfall peak impact force, the ratio of peak impact force to peak interaction force, and lateral displacement of pile at the ground surface had strong linear relationships with rockfall energy.

Academic contribution: The paper increases the knowledge in civil engineering. The paper deserved to be published in *NHESS*.

Figures Figures are of acceptable quality, easy to read, relevant and suitable.

Recommendation: This manuscript can be accepted based on the detailed report above.

With kind regards,

- Anonymous Reviewer.

10.04.2024.

---

## Author Response (AR2)

Dear editors:

We gratefully thank for your constructive remarks and useful suggestions, which has significantly raised the quality of the manuscript and facilitated its improvement. Below the comments of the reviewers are responses point by point and the revisions are indicated.

**Reviewer**

**1. General Comments:**

Rewrite the abstract so that it reflects the context, methods, main findings and conclusions of the paper.

**1. Reply**

We gratefully appreciate for your valuable suggestion. A rewritten abstract is as follows:

**Abstract:** The pile-slab retaining wall, as an innovative rockfall protection structure, has been extensively utilized in the western mountainous regions of China. With its characteristics of a small footprint, high interception height, and ease of construction, this structure demonstrates promising potential for application in mountainous regions worldwide, such as the Himalayas, Andes, and Alps. However, its dynamic response upon impact and impact resistance energy remain ambiguous, due to the intricate composite nature of the structure. To elucidate this, an exhaustive dynamic analysis of a four-span pile-slab retaining wall with a cantilever section of 6 m under various impact scenarios was conducted utilizing the finite element numerical simulation method. The rationality of the selected material constitutive models and the numerical algorithm was validated by reproducing two physical model tests. The simulation results reveal the following: (1) The lateral displacement of the pile at the ground surface and the concrete damage under the pile as the impact center is greater than those under the slab as the impact center, implying that the impact location has a significant influence on the stability of the structure. (2) There is a positive correlation between the response indexes (impact force, interaction force, lateral deformation of pile and slab, concrete damage) and the impact velocities. (3) The rockfall peak

impact force, the ratio of peak impact force to peak interaction force, and lateral displacement of pile at the ground surface had strong linear relationships with rockfall energy. (4) Relative to the bending moment, shear force and damage degree, the lateral displacement of pile at the ground surface is the first to reach its limit value. Taking the lateral displacement of the pile at the ground surface as the controlling factor, the estimated maximum impact energy that the pile-slab retaining wall can withstand is 905 kJ in this study when the structure top is taken as the impact point. In cases where the impact energy of falling rocks exceeds 905 kJ, it is recommended to optimize the mechanical properties of the cushion layer, improve the elastic modulus of concrete, increase the reinforcement ratio of longitudinal tension bars, enlarge the section size of pile at ground level, or add anchoring measures to enhance the bending resistance of the retaining structure.

**2. General Comments:**

Highlight the novelty of the own studies. This can be done by highlighting the research questions, the research hypothesis, the assumptions and the limitations. The conclusions shall repeat the initial claim in the light of the conclusions which have been reached in the paper regarding the research hypothesis/the thesises. This way either the conclusion or the introduction will repeat the abstract.

**2. Reply**

Thanks for your comment. We made a serious We carefully reviewed the whole paper and rewrote the conclusion. A rewritten conclusion is shown below:

Compared to existing rockfall protection structures, the PSRW offers enhanced stability and requires a smaller footprint, making it adept at addressing a broad spectrum of    rockfall impact scenarios commonly encountered in alpine canyon regions. In this paper, the dynamic response of the PSRW under different impact centers and velocities were compared and analyzed using the FEM simulation method. Additionally, the influencing factors such as peak impact force, peak interaction force, ratio of peak impact force to peak interaction force, concrete stress, reinforcement stress, maximum lateral displacement of the pile at the ground surface, and ratio of

damage failure units to overall structure units were quantified. Notably, the formula for calculating the peak impact force of the PSRW (Eqs. 1), the ratio of peak impact force to peak interaction force (Eqs. 2), maximum lateral displacement of the pile at the ground surface (Eqs. 3) based on the impact energy of rockfalls were proposed. The key findings of this study are as follows:

(1) The impact force, interaction force and lateral displacement exhibit a linear correlation with the impact velocity. however, the lateral displacement is more sensitive to velocity variations than the impact force and interaction force.

(2) Under different impact centers, the variations in impact force and interaction force are minimal. When the pile serves as the impact center, the lateral displacement of pile at the ground surface and the extent of concrete damage are significantly greater than when the slab center is the impact center. This indicates that impacts centered on the pile pose a more hazardous impact scenario.

(3) Concrete damage predominantly concentrates at the joints between piles and slabs, the impact center itself, and the section of piles at the ground surface. To minimize structural concrete damage, it is imperative to prioritize these critical sections in the structural design.

(4) The impact force, the ratio of peak impact force to peak interaction force, and the maximum lateral displacement of the pile at the ground surface have a significant correlation with the impact energy. These relationships are crucial for evaluating impact force, interaction force, and the lateral displacement of piles at ground surface during the design of PRSW structures. According to Chinese specifications for displacement requirements, the maximum lateral displacement of the pile at the ground surface should not exceed 10 mm. Consequently, the maximum impact energy that the PSRW can withstand is 905 kJ, when the crown is designated as the impact center.

**3. General Comments**

A more practical relationship between the concrete (site) study and the mathematical model shall be reached.

**3. Reply**

We gratefully appreciate for your valuable suggestion. A rewritten part about the practical relationship between the concrete (site) study and the mathematical model is shown below:

The design drawing of the PSRW (Fig. 3) is consistent with the actual project located in Zhangmu Town, China. Given the large scale of the actual engineering structure, numerical simulations have been focused solely on a representative four-span structure, incorporating appropriately simplified boundary conditions to facilitate the analysis. For a comprehensive understanding of the modeling specifics, kindly refer to Section 2.1.3.

---

## Author Response (AR3)

Dear Reviewer:

We gratefully thank you for your constructive remarks and useful suggestions, which have significantly improved the quality of the manuscript and have enabled us to refine it. Below are our responses to the reviewers' comments, point by point, and the revisions are indicated.

**Reviewer**

**General Comments:**

However, the research question connected to the PSRW (pile slab retaining wall) may be better highlighted in the introduction, by placing it in a separate paragraph, together with the figures.

**Reply:**

We are deeply grateful for your invaluable suggestions. After meticulous consideration, we have revised the capacity of this section. The revised content is presented below:

In response to the challenges posed by steep terrains, narrow site conditions, and suboptimal foundation conditions in mountainous terrain, Hu et al. (2019) introduced the PSRW structure. This structure comprises a buffer layer and an anti-slide pile-slab system, which has found widespread application in southwestern China (Fig. 1). Due to the use of pile foundations, this structure exhibits characteristics such as a small footprint, high interception height, and ease of construction.

[Figure]

Fig. 1. PSRW in south-western China (a) Kongyu town (b) Jiuzhaigou nature reserve (c) Zhenjiangguan tunnel exit in Chengdu-lanzhou railway (d) Zhangmu town

However, the current PSRW design verification approach treats the structure as either an underground continuous wall (CAGHP, 2019) or an elastic cantilever beam (Tian et al., 2024). The structural design primarily considers the impact force of falling rocks as the sole external influencing parameter, while the impact energy is seldom taken into account. Furthermore, existing research primarily focuses on single slabs and piles impacted by rockfall (Wu et al., 2021; Yong et al., 2021). Consequently, due to the scarcity of comprehensive research reports on the ultimate load-bearing capacity of this structure, it is frequently overlooked during the initial selection of protective structures, and potential failure scenarios may be underestimated (Fig. 2). Additionally, because of the composite nature of this structure, the dynamical response at various impact points remains elusive.

[Figure]

Fig. 2. Some PSRW in Zhangmu Town were damaged by falling rocks.